# Variant Allele of *ALDH2*, rs671, Associates with Attenuated Post-Vaccination Response in Anti-SARS-CoV-2 Spike Protein IgG: A Prospective Study in the Japanese General Population

**DOI:** 10.3390/vaccines10071035

**Published:** 2022-06-28

**Authors:** Akiko Matsumoto, Megumi Hara, Mohammad Said Ashenagar, Mikiko Tokiya, Takeshi Sawada, Chiharu Iwasaka, Takuma Furukawa, Kyoko Kitagawa, Yasunobu Miyake, Yoshio Hirota

**Affiliations:** 1Department of Social and Environmental Medicine, School of Medicine, Saga University, 5-1-1 Nabeshima, Saga 849-8501, Japan; sx5080@cc.saga-u.ac.jp (M.S.A.); sx4932@cc.saga-u.ac.jp (M.T.); 2Department of Preventive Medicine, Faculty of Medicine, Saga University, 5-1-1 Nabeshima, Saga 849-8501, Japan; harameg@cc.saga-u.ac.jp (M.H.); sy7509@cc.saga-u.ac.jp (C.I.); sr0753@cc.saga-u.ac.jp (T.F.); 3Division of Histology and Neuroanatomy, Department of Anatomy and Physiology, Faculty of Medicine, Saga University, 5-1-1 Nabeshima, Saga 849-8501, Japan; 22624004@edu.cc.saga-u.ac.jp; 4Department of Environmental Health, University of Occupational and Environmental Health, 1-1 Iseigaoka, Yahatanishi-ku, Kitakyushu 807-8555, Japan; kitakyo@med.uoeh-u.ac.jp; 5Division of Molecular and Cellular Immunoscience, Department of Biomolecular Sciences, Faculty of Medicine, Saga University, Saga 840-8501, Japan; ymiyake@cc.saga-u.ac.jp; 6SOUSEIKAI Medical Group, Clinical Epidemiology Research Center, Medical Co., LTA, 3-6-1 Kashii-Teriha, Higashi-Ku, Fukuoka 813-0017, Japan; hiro8yoshi@lta-med.com

**Keywords:** ALDH2, rs671, COVID-19, vaccine, immunogenicity

## Abstract

Uncovering the predictors of vaccine immunogenicity is essential for infection control. We have reported that the most prevalent polymorphism of the aldehyde dehydrogenase 2 gene (*ALDH2*), rs671, may be associated with an attenuated immune system. To test the inverse relationship between rs671 and antibody production after COVID-19 vaccination, the levels of anti-SARS-CoV-2 Spike protein S1 subunit (S1) IgG were repeatedly measured for four months before and after vaccination with BNT162b2 or mRNA-1273, in 88 Japanese workers and students (including 45 females, aged 21–56 years, with an rs671 variant allele frequency of 0.3). The mixed model including fixed effects of the vaccine type, weeks post vaccination (categorical variable), sex, age, height, smoking status, ethanol intake, exercise habit, perceived stress, steroid use, allergic diseases, and dyslipidemia, indicated an inverse association between log-transformed anti-S1 IgG levels and the number of rs671 variant alleles (partial regression coefficient = −0.15, *p* = 0.002). Our study indicated for the first time that the variant allele of *ALDH2*, rs671, is associated with the attenuated immunogenicity of COVID-19 mRNA vaccines. Our finding may provide a basis for personalized disease prevention based on a genetic polymorphism that is prevalent among East Asians.

## 1. Introduction

Aldehyde dehydrogenase 2 (ALDH2), a member of the ALDH superfamily [1], is expressed in most human tissues, including immune cells, and is crucial for the metabolism of endogenous aldehydes, such as formaldehyde and 4-hydroxynonenal [2]. The rs671 polymorphism, which results from missense mutations in the coding region of the *ALDH2* gene, is the most common ALDH2 deficiency in humans, exclusively observed in the East Asian population, with an incidence of 40–50% in certain populations, such as Japanese, Taiwanese, and Han Chinese [3,4,5] (https://www.ncbi.nlm.nih.gov/snp/rs671, accessed on 21 June 2022), thereby accounting for 5–10% of the world population. Although the rs671 polymorphism has never been reported to be associated with vaccine efficacy, it is reported to be involved in various traits, including lifestyle habits, disease risks, and drug sensitivities [2,6,7,8,9]. Aiming for a proposal of personalized medicine based on rs671, we have performed several investigations on its unique and novel phenotypes [10,11], including inhibited T cell immunity [12]. Considering intercellular communication in the immune system, B cells and antibody production may also be affected.

The immune system defends the body against infections from various pathogens, including viruses. In the context of the COVID-19 pandemic, it is well known that the extent of the infection, as well as the vaccine efficacy, varies across individuals. This suggests the need for research on the various factors that affect vaccine immunogenicity, both genetic as well as those pertaining to lifestyle. Numerous studies have indicated that the vaccine efficacy depends on several factors, such as the type of vaccine, number of doses, and the demographical and clinical characteristics of recipients [13,14,15,16,17]. Although antibody responses against SARS-CoV-2 are characterized by responses against a range of viral proteins, including spike proteins, nucleoproteins, and membrane proteins, the T cell response is a critical component of immune protection against SARS-CoV-2 [18,19,20]; T cell responses to these proteins are reportedly correlated with the antibody levels [21,22,23]. These findings suggest a relationship between rs671 and antibody production.

Therefore, the present study aimed to investigate the immune response in a Japanese population, before and after the administration of the COVID-19 vaccination, with the hypothesis that there is an inverse relationship between *ALDH2* rs671 and antibody production.

## 2. Materials and Methods

This study was approved by the Ethics Committee for Clinical Research of the School of Medicine Saga University, Saga, Japan (No. R2-44 and R3-9). All participants provided written informed consent before undergoing any study procedure.

### 2.1. Study Design and Participants

The study group comprised 88 participants from hospitals and a university in Saga prefecture, who were invited to be vaccinated with two mRNA vaccines: 62 participants (20 healthcare workers and 42 students) with two doses of BNT162b2 (Pfizer Inc., New York, NY, USA/BioNTech SE, Mainz, Germany) (30 µg) and 26 participants (26 university employees and students) with mRNA-1273 (Moderna Inc., Cambridge, MA, USA/Takeda Pharmaceutical Co., Ltd., Tokyo, Japan) (100 µg). The first dose was scheduled for April and May 2021, and the second dose was administered 21 and 28 days after the first dose for BNT162b2 and Moderna-mRNA-1273, respectively. None of the participants had a history of COVID-19 infection.

### 2.2. Serological Tests

Blood samples were collected before the first vaccination and every other week after the second vaccination for healthcare workers; likewise, samples were collected before the first dose, three weeks after the first vaccination, and four weeks after the second vaccination for the university employees and students. Serum was extracted from the samples on the same day and stored at −80 °C until analysis. A high-sensitivity chemiluminescent enzyme immunoassay (CLEIA) platform (Sysmex Co., Kobe, Japan) was used to measure the three anti-SARS-CoV-2 antibodies, the S1 subunit of the anti-spike protein (S1) IgG, anti-S1 IgM, and anti-nucleocapsid protein (N) IgG [24]. The unit for anti-S1 IgG, IgM, and ant-N IgG is binding antibody units per mL (BAU/mL), Sysmex unit per mL (SU/mL), and SU/mL, respectively. BAU was calibrated using the WHO International Standard.

### 2.3. Self-Administered Questionnaire

A self-administered questionnaire was employed to ask about sex, age, height, weight, smoking status, alcohol intake, exercise habit, perceived stress, and medical history. A positive smoking status was defined as cigarette smoking at the time of the application of the questionnaire. None of the participants had changed their smoking habits in the preceding year. Ethanol intake was calculated based on the amount of alcohol consumed in the previous six months, adjusted per 60 kg of body weight, and then categorized into <1 g/day, ≥1 g/day, <20 g/day, and ≥20 g/day. Exercise habit was assessed by asking, “Do you usually exercise?”, with possible answers including, no habit, <1 day/week, 1 to 3 days/week, and ≥3 days/week. The question “Do you feel psychological stress?” was asked to evaluate perceived stress on a 5-point scale, no (0), mostly no (1), unsure (2), quite often (3), and yes (4). Steroid use was considered as “yes” if the participants were receiving steroids at the time; none of the participants who answered “no” had received steroids in the preceding 3 years. The allergic disease condition was assessed with the question, “Do you have allergic diseases?”. Dyslipidemia was considered as “yes” if the participants had concurrent dyslipidemia; those who answered “no” had no history of the disease in the 3 years prior.

#### 2.3.1. Covariates

Alongside sex, age, vaccine type, and weeks post vaccination, we included height, smoking status, ethanol intake, exercise, perceived stress, steroid use, allergic disease, and dyslipidemia as covariate attributes suspected to be associated with vaccine efficacy, immune response, or rs671 [13,25,26,27,28,29,30].

#### 2.3.2. Sensitivity Analysis

Out of the 88 subjects, one participant with dyslipidemia was excluded for the sensitivity analysis (87 subjects, 493 data points). We also performed additional analysis using body weight or log-transformed body mass index instead of height. The number of observations resulted in 499 data points for the 87 subjects, because the body weight was unknown for one of the participants.

### 2.4. Genotyping

The *ALDH2* genotype (rs671) was determined using the DNA extracted from blood clots as follows: Approximately 0.1 mL of blood clots was incubated in 0.4 mL of proteinase K solution (proteinase K at 1–10 U/mL in 0.01 M Tris-HCl, pH 8 with 0.01 M EDTA and 0.5% sodium dodecyl sulfate) at 56 °C overnight, then 0.5 mL of TE-saturated phenol was added. After vigorous mixing for 20 s, the samples were incubated on ice for 10 min, followed by centrifugation at 16,000× *g* for 5 min at room temperature (20–25 °C). The aqueous layer was separated, and 0.5 mL of ethanol (95–100%) was added to it, mixed well, and then incubated at room temperature for 10 min. After centrifugation at 12,000 rpm for 10 min, the precipitated DNA was collected by discarding the supernatant. The DNA pellets were washed with 0.25 mL of 70% ethanol, dried, and dissolved in 20–200 µL DNase-free water. The DNA samples were then genotyped using a TaqMan^®^ SNP genotyping assay system following the manufacturer’s instructions (ThermoFisher Scientific, Waltham, MA, USA).

### 2.5. Statistical Analyses

Mixed models were used to compute the association between the rs671 genotype and the log-transformed antibody levels to account for repeated measurements and the random effect of the subpopulation (proc mixed by SAS9.4 TS Level 1M5 for Windows, SAS Institute, Cary, NC, USA). Statistical significance was set at *p* < 0.05. To verify the assumption of the additive effect of the rs671 variant allele, the least squares geometric means and standard errors were computed for each genotype, using mixed models that included the interactive terms of weeks post vaccination (categorical) and rs671 genotype (categorical), and graphically presented. In the figures, the *x*-axis is represented on a log 2 scale because preliminary computing showed that for “weeks” the B-spline regression fits better using the logarithm rather than its antilogarithm (proc hpmixed with B-spline effect, where Y = log(IgG) and X = log(week) or X = week, SAS9.4).

## 3. Results

### 3.1. Baseline Characteristics

Table 1 shows baseline characteristics for the 88 participants, including 45 (51%) females, aged 21–56 years, with the confirmation of the rs671 genotype, namely the wild-type homozygous, *ALDH2*1/*1* (GG-type, N = 44), heterozygous, *ALDH2*1/*2* (GA-type, N = 33), and variant homozygous, *ALDH2*2/*2* (AA-type, N = 11). The variant allele frequency was 0.313, and the genotype frequency did not differ from that expected from Hardy–Weinberg equilibrium (*p* = 0.5 by χ^2^ test). Daily ethanol intake (g/day) was low in the variant allele carriers (GA- and AA-types); medians and interquartile ranges were 0.4 (0.1–1.2), 0.06 (0–0.41), and 0 (0–0) for GG-, GA-, and AA-types, respectively (*p* < 0.0001 by Spearman rank correlation). The distribution of exercise habits, perceived stress, and allergic disease was not different among the three groups (*p* > 0.4 by Fisher’s exact test). Steroid use was reported only by two participants of the GG-type, and dyslipidemia by one participant of the AA-type.

### 3.2. Antibody Production Post Vaccination

Anti-N IgG levels were below 0.7 SU/mL during the entire observation period for all participants. The medians and respective interquartile ranges of the anti-S1 IgG levels are shown in Table 2. Antibody titers measured in the third week for all participants were distributed in the range of 19–409 BAU/mL in the BNT162b2 group and 46–1369 BAU/mL in the mRNA-1273 group. In the healthcare worker group, antibody titers peaked in the fifth week (two weeks after the second dose) and ranged from 592 to 7895 BAU/mL.

### 3.3. Effect of rs671 on Anti-S1 IgG Post Vaccination

The fixed effects of baseline characteristics that could explain log-transformed anti-S1 IgG, repeatedly measured for 88 subjects, are shown in Table 3. In model 1, which included baseline characteristics, vaccine type, sex, and age, the partial regression coefficient (β) was estimated to be −0.11 (*p* = 0.01). The effect of the rs671 allele was estimated to be stronger when height, lifestyle habits other than ethanol intake, and current medical history were included in the models (β = −0.13, *p* = 0.002 in model 2). Further adjustment for alcohol intake suggested an even stronger association, with a β value of −0.15 (*p* = 0.002 in model 3). The vaccine type had a strong effect on antibody levels, and no interactive effect between rs671 and the type of vaccine was indicated (*p* for interaction = 0.5 in a model additionally includes the interactive term to model 3).

Sensitivity analysis, excluding one participant who had dyslipidemia, resulted in a similar estimation (N = 493, AIC = 976, β = −0.15, *p* = 0.002 in model 3). A slightly modified model 3, including weight instead of height, produced similar estimates (N = 499, AIC = 989, β = −0.16, *p* = 0.001), similar to the model using log-transformed body mass index instead of height (N = 499, AIC = 982, β = −0.16, *p* = 0.001). Anti-S1 IgM was not associated with rs671 (Appendix A).

The least squares geometric means computed using a mixed model depicted the adjusted correlation between the post-vaccination period and the rs671 genotype (Figure 1 and Appendix A). Anti-S1 IgG levels differed most significantly 1–3 weeks after administration of the second dose (Figure 1). The adjusted geometric means of IgG levels 2 weeks after administration of the second dose were estimated to be 3090, 1843, and 1098 BAU/mL for the participants carrying the GG-, GA-, and AA-type alleles, respectively. No such association was found for the IgM levels (Appendix A).

## 4. Discussion

Our study is the first to show the effect of the *ALDH2* polymorphism, rs671, which is carried by nearly half of all East Asians, on vaccine immunogenicity, while no association was found between rs671 and anti-S1 IgM levels. The effect of genetic polymorphisms on immunity after COVID-19 has been well-reported in limited types of genes encoding proteins directly related to the immune response to SARS-CoV-2, such as *ACE* and *HLA* [31,32]. However, to the best of our knowledge, genetic polymorphisms such as rs671, which may strongly affect large populations, have never been reported.

We found the largest difference 2 weeks after the second vaccination; the adjusted geometric means of anti-S1 IgG of the GG-type were almost 3-fold those of the AA-type. However, the difference became smaller, e.g., 706 and 467 BAU/mL for GG- and AA-type, respectively, 3 months after the second vaccination. Thus, the effect of rs671 on the immunogenicity of the COVID-19 mRNA vaccine in general populations may have limited its clinical significance. However, it can be relevant in populations with weak immune responses. Considering the vaccine-hesitancy phenomenon owing to the strong side effects of these vaccines [33], it may be necessary to optimize the number of doses and timing of administration, and in this case, rs671 may become an important factor to consider, even in general populations.

In our study, we included multiple covariates, such as smoking status, ethanol intake, exercise, perceived stress, steroid use, allergic disease, and dyslipidemia, to improve the validity and rigor of our statistical assumption based on previous research. Previous studies indicated glucocorticoids, allergy, and ethanol consumption to be associated with vaccine efficacy [13], whereof glucocorticoid is a marker of perceived stress [25]. Exercise is known to affect the immune system [26]. Statin, an effective lipid-modifying drug, is suspected to be associated with the clinical outcomes of COVID-19 [27,28]. Additionally, rs671 is reportedly associated with cigarette smoking behavior [29,30], with a strong effect on drinking behavior, as confirmed in our cohort.

The results of this study should be interpreted with the careful consideration of a few limitations. Most importantly, the number of subjects used in the study was low. In particular, the estimated IgG levels at weeks 4, 5, and 6 post vaccination (1−3 weeks after administration of the second dose), which were most affected by rs671, were based on the measurements derived from health workers (20 subjects). Furthermore, the number of participants that carried homozygous variants was small (3 health workers and 11 participants overall); we tried to address this issue by computing a linear regression between the number of variant alleles and the log-transformed IgG values. Another important limitation is the generalizability of this study. Personalizing medication strategies is more important in immunogenically disadvantaged populations, such as patients with autoimmune diseases or those on medication with immunosuppressive drugs [23]; however, this study only included healthy Japanese students and workers.

Several possible mechanisms exist for the inhibition of elevation in IgG levels via rs671. First, since our previous study suggested the suppression of the T-cell mediated immune response in patients with thoracic malignancies carrying the rs671 variant allele [11], it can be speculated that the current findings are a consequence of the functional disturbance in CD4+ T cells. The absence of such an effect on anti-S1 IgM is consistent with this speculation because IgG is produced by class-switched memory B cells, which are triggered by cytokines released by CD4+ T cells, while IgM production is independent of this signaling. Moreover, we reported that the rs671 variant allele and CD4+ T cell count are inversely related in Japanese workers (N = 328 with 48% of males, *p* trend = 0.07, by a generalized linear model including covariates of sex, age, year of survey, alcohol consumption, rs671, and the interactive term of alcohol consumption*rs671) (conference proceeding) [12]. Therefore, the association between the anti-S1 IgG and rs671 could be attributed to the low CD4+ T cell count, which requires further examination of the hypothesis for different vaccines. Second, Brunsdon et al. (2022) reported that ALDH2 deficiency delays melanocyte differentiation owing to a deficiency in endogenous formic acid (a metabolite of endogenous formaldehyde), which is required for nucleic acid synthesis [34]. Furthermore, progenitor hematocytes express ALDH [35], and evidence indicates the dependency of those cells on ALDH2 among other ALDH isozymes [36]. These findings indicate that ALDH2 deficiency inhibits the differentiation of naïve T cells to effector T cells, which promotes the class-switching of B cells, consequently accelerating the increase in IgG production.

## 5. Conclusions

Our study indicated for the first time that the variant allele of *ALDH2*, rs671, which is prevalent in East Asians, is associated with the attenuated immunogenicity of the COVID-19 mRNA vaccine, especially after administration of the second dose. However, further epidemiological investigations and experimental approaches are necessary to confirm this hypothesis and elucidate the mechanistic details of the effects of this allele on CD4+ T cell functioning. This finding may provide evidence for personalized medicine based on a common genetic polymorphism, in addition to promoting a basic understanding of immunology. While this study focuses on healthy individuals, further research on the immune responses in larger, more varied populations, including subjects with prior health conditions, could lead to a better understanding of vaccine immunogenicity.

## Figures and Tables

**Figure 1 vaccines-10-01035-f001:**
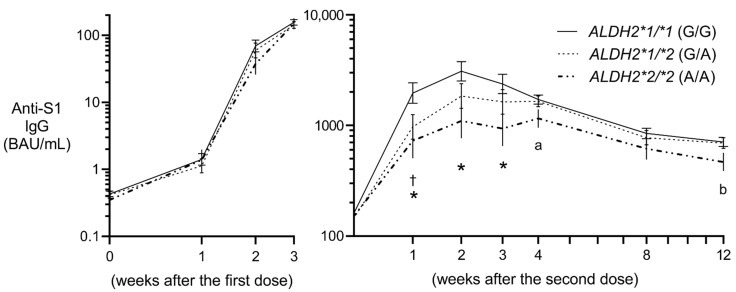
Estimated anti-S1 IgG antibody levels by *ALDH2* rs671 genotype. Least squares geometric means and standard errors were computed by a mixed model, which includes all the covariates presented in Table 1, vaccine type, the number of weeks (categorical variables), rs671 genotype, and the interactive terms, rs671*the number of weeks, as fixed effects, and random effects by repeated measures and by the three cohorts. BAU, binding antibody units, calibrated using the WHO international standard. The *x*-axis is represented on log 2 scale. * *p* < 0.05, ^a^ *p* = 0.067, ^b^ *p* = 0.053 for the comparisons between *ALDH2*1/*1* (GG) and *ALDH2*2/*2* (AA). ^†^ *p* < 0.05 for the comparison between *ALDH2*1/*1* (GG) and *ALDH2*1/*2* (GA).

**Table 1 vaccines-10-01035-t001:** Baseline characteristics of participants for *ALDH2* rs671 polymorphism.

Participants	Healthcare Workers			University Students			University Employees and Students		
N	20			42			26		
First Dose	April 2021			May 2021			May 2021		
Type of Vaccine	BNT162b2			BNT162b2			mRNA-1273		
Second Dose	Three Weeks after the First Dose			Three Weeks after the First Dose			Four Weeks after the First Dose		
	GG	GA	AA	GG	GA	AA	GG	GA	AA
Males, N	3	4	3	11	6	5	3	7	1
Females, N	7	3	0	13	7	0	7	6	2
Age, years									
Median	42	36	35	22	22	22	39	47	21
(IQR)	(36–49)	(25–43)	(25–40)	(22–23)	(22–23)	(22–23)	(22–56)	(21–55)	(21–42)
Body height, cm									
Median	164	168	170	165.5	162	172	162	166	157
(IQR)	(162–169)	(153.8–178)	(170–176)	(158–170.5)	(156–174)	(170–173)	(155–166)	(163–170)	(151–163)
Smoking status, yes	1	1	1	0	0	0	0	1	0
Ethanol intake *									
<1 g/d	1	4	3	12	8	4	6	9	3
≥1, <20 g/d	7	3	0	12	5	1	4	2	0
≥20 g/d	2	0	0	0	0	0	0	2	0
Exercise habit									
No habit	8	2	1	10	4	2	3	4	1
<1 d/w	0	0	1	5	1	1	2	2	0
1 to 3 d/w	2	2	1	5	6	2	2	4	1
≥3 d/w	0	3	0	4	2	0	3	3	1
Perceived stress									
0 (no)	2	3	1	12	5	1	1	5	1
1	0	0	1	2	1	0	3	1	0
2	3	1	1	4	4	3	2	2	0
3	4	2	0	6	3	1	4	4	0
4 (yes)	1	1	0	0	0	0	0	1	2
Steroid use, yes	0	0	0	2	0	0	0	0	0
Allergic disease, yes	2	4	0	11	3	1	2	6	1
Dyslipidemia	0	0	1	0	0	0	0	0	0

GG, GA, and AA represent the genotypes of rs671, i.e., *ALDH2*1/*1*, *ALDH2*1/*2*, and *ALDH2*2/*2*, respectively. IQR, interquartile range. * Ethanol intake was adjusted for body weight (g/day/60 kg body weight).

**Table 2 vaccines-10-01035-t002:** Anti-S1 IgG levels (BAU/mL) according to *ALDH2* rs671 genotype.

	Healthcare Workers			University Students			University Employees and Students		
	BNT162b2 (Second Dose at Week 3)			BNT162b2 (Second Dose at Week 3)			mRNA-1273 (Second Dose at Week 4)		
Genotype	GG	GA	AA	GG	GA	AA	GG	GA	AA
**Week 0**									
N	10	7	3	24	13	5	10	13	3
Median	0.37	0.49	0.41	0.39	0.46	0.35	0.49	0.46	0.34
(IQR)	(0.33–0.5)	(0.45–0.58)	(0.4–0.53)	(0.33–0.52)	(0.35–0.57)	(0.28–0.39)	(0.3–0.64)	(0.41–0.52)	(0.3–0.62)
**Week 1**	1 week after the first dose								
N	10	7	3						
Median	0.62	0.54	0.5						
(IQR)	(0.37–1.76)	(0.51–1.15)	(0.37–7.2)						
**Week 2**	2 weeks after the first dose								
N	10	6	3						
Median	64	46	50						
(IQR)	(44–89)	(18–76)	(5.71–95)						
**Week 3**	3 weeks after the first dose			3 weeks after the first dose			3 weeks after the first dose		
N	10	6	3	24	13	5	10	13	3
Median	187	92	122	132	134	122	372	214	560
(IQR)	(101–295)	(70–113)	(56–153)	(75–210)	(91–200)	(105–135)	(290–393)	(143–320)	(410–572)
**Week 4**	1 week after the second dose								
N	9	6	3						
Median	1498	629	898						
(IQR)	(691–2529)	(576–1271)	(241–909)						
**Week 5**	2 weeks after the second dose								
N	10	6	3						
Median	2482	1694	725						
(IQR)	(1901–2667)	(956–2523)	(592–1594)						
**Week 6**	3 weeks after the second dose								
N	10	6	3						
Median	1958	1372	564						
(IQR)	(1507–2113)	(918–1975)	(558–1350)						
**Week 7**	1 month after the second dose			1 month after the second dose					
N	10	6	3	24	13	5			
Median	1592	1109	510	1597	1880	1339			
(IQR)	(1129–1779)	(619–1658)	(433–818)	(1198–2269)	(1461–2292)	(1174–1680)			
**Week 8**							1 month after the second dose		
N							10	13	3
Median							3200	2959	2854
(IQR)							(2756–3681)	(1661–3593)	(1362–3565)
**Week 11**	2 months after the second dose			2 months after the second dose					
N	10	6	3	24	12	5			
Median	761	526	337	940	859	632			
(IQR)	(493–854)	(419–819)	(228–523)	(560–1080)	(534–1133)	(572–752)			
**Week 15**	3 months after the second dose			3 months after the second dose					
N	10	6	3	24	12	5			
Median	361	341	162	815	680	426			
(IQR)	(271–486)	(300–402)	(113–331)	(464–1087)	(457–1085)	(418–677)			
**Week 16**							3 months after the second dose		
N							10	13	3
Median							1579	1227	1268
(IQR)							(1220–1740)	(742–1906)	(983–1746)

BAU, binding antibody units, calibrated using the WHO International Standard. GG, GA, and AA represent the genotypes of rs671, i.e., *ALDH2*1/*1*, *ALDH2*1/*2*, and *ALDH2*2/*2*, respectively. IQR, interquartile range.

**Table 3 vaccines-10-01035-t003:** Estimated fixed effects of baseline characteristics on log-transformed anti-S1 IgG, BAU/mL.

	Model 1		Model 2		Model 3	
	AIC = 1007		AIC = 993		AIC = 997	
	503 Observations		503 Observations		503 Observations	
	88 Subjects		88 Subjects		88 Subjects	
Fixed Effects	β	*p*-Value	β	*p*-Value	β	*p*-Value
BNT162b2 (reference)						
mRNA-1273	0.53	0.0004	0.50	<0.0001	0.48	<0.0001
Age (per year old)	−0.01	0.0087	−0.01	0.0007	−0.01	0.0030
Female sex	0.21	0.0013	0.07	0.4522	0.05	0.6345
Height (per cm)			−0.01	0.2000	−0.01	0.1725
Smoking status, yes			0.20	0.1138	0.20	0.1039
Ethanol intake (per category)					−0.05	0.3908
Exercise habit (per category)			−0.03	0.2716	−0.03	0.2802
Perceived stress (per category)			0.07	0.0043	0.07	0.0041
Steroid use, yes			−0.07	0.7518	−0.09	0.6877
Allergic disease, yes			−0.04	0.5208	−0.04	0.4938
Dyslipidemia, yes			−1.02	<0.0001	−1.02	<0.0001
*ALDH2* variant allele number	−0.11	0.0116	−0.13	0.0021	−0.15	0.0016

The effects of baseline characteristics were computed using mixed models to account for repeated measurements and the random effect of the subpopulation. All models include the fixed effects of post-vaccination week as a categorical variable and variables listed in the table. BAU, binding antibody units, calibrated using the WHO International Standard. β, partial correlation coefficient.

## Data Availability

The data presented in this study are available on request from the corresponding author (A.M.). The data are not publicly available owing to privacy concerns.

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
