# Peer review of "Variant Allele of ALDH2, rs671, Associates with Attenuated Post-Vaccination Response in Anti-SARS-CoV-2 Spike Protein IgG: A Prospective Study in the Japanese General Population"

_vaccines, 2022, doi:10.3390/vaccines10071035_

Round 1

Reviewer 1 Report

The manuscript is scientifically sound, nicely written, shedds some insight on the reasons of post-vaccination immunity differences that are interesting to the readers for practical reasons as well.

Major concern - as I've concluded from the data presented in the manuscript the differences in antibody levels between the vaccinees with different genetic alleles are significant only short-term after vaccination, and the differences diminish with time. I think it should be highlighted in Conclusion and in the Title as well.

Minor concern - the tables are very hard to read - if it is possible the IQR in the brackets should fit 1 line. 

Author Response

Thank you very much for your time and comments.

Although we failed to mention it in the original manuscript, there are borderline differences between the GG and AA genotypes even in the late phase of our observation. So we modified Fig 1 to make it easier to understand the borderline differences. The smaller differences in the late phase are discussed in the second paragraph in Discussion in the revised manuscript.

Regarding the table, thank you for the comment, we tried to make them easier to see.

Reviewer 2 Report

The manuscript performs a retrospective study by measuring the level of IgG/IgM and its association with the allele variant of ALDH2 (rs671), distributed in an Asian population. The work employs a set of statistical tools to reach correlation conclusions. The subject is of interest, and although the number of the individuals and controls is low and a great deal of speculation is presented in the discussion, we consider that the manuscript deserves publication due to its relevance. First, however, it is necessary that some points are better described and clarified. Also a review of English is necessary.

major points

Line 52- “anti-body”- correct for antibody

Line 158/Table 2- “What does SU/ml and BAU/ml mean?”- Please specify at the table footer.

Line 169- “partial regression coefficient (β) was estimated to be -0.11 (p = 0.01), which represents a 20% reduction in the IgG levels of the AA-type”- This statement needs to be further explained and discussed. How did the authors arrive at this number of reduction of specific anti-S1 antibodies? Does the allele control only the anti-COVID-19 level or the entire IgG and/or other Ig humoral immune response? See that no correlation was found for IgM. Please explain the results better. A variation of the Ig level of only 20% can be considered significant in a group of patients? Why did the analysis not consider each patient individually?

Line 174-176- “Although the vaccine type had a strong effect on antibody levels, no interactive effect between rs671 and the type of vaccine was indicated (p for interaction = 0.5 in a model additionally includes the interactive term to model 3)”. As no correlation was found if there is a difference in the level of antibodies between the 2 vaccines, as is known and stated in the sentence itself?

Line 209- “In our study, we included multiple covariates…”- Include what these covariates were in a parenthesis.

Author Response

Thank you very much for your time and comments.

We corrected “anti-body” to antibody, and other similar mistyping, and also specified SU and BAU in the footnotes and figure legends.

Regarding the "20% reduction" issue, we reconsider the statements and concluded to delete them.  ”20% reduction” was simply calculated by beta value, however, because this is on the assumption that rs671 has the same effect in the entire observation period, this number is almost meaningless. The size of the difference (reduction) is better presented in modified Fig 1.

Regarding the significance, we agreed and discussed in the second paragraph of Discussion. Basically, we believe that the clinical relevance our study suggested is limited to a healthy population in terms of vaccine efficacy, however, it would be an important suggestion for special populations with weak immunity, because the overlapping risk factor is often crucial (for example, the Fanconi anemia gene polymorphism and rs671 overlapping turn out to a severe prognosis).  

No difference in IgM is in line with our hypothesis, because different levels of IgM do not fit the hypothesis that class switching by CD4-positive T cells is affecting (this study was designed because we found the effect of rs671 on CD4 and CD8 T cell function). Therefore, as you are concerned, this should be consistently found for different kinds of vaccines, not only the COVID-19 vaccine. We referred to it in the last paragraph of the Discussion. 

Regarding the following comment:

"Line 174-176- “Although the vaccine type had a strong effect on antibody levels, no interactive effect between rs671 and the type of vaccine was indicated (p for interaction = 0.5 in a model additionally includes the interactive term to model 3)”. As no correlation was found if there is a difference in the level of antibodies between the 2 vaccines, as is known and stated in the sentence itself?"

we are sorry for the confusing English, we meant no interaction between rs671 and vaccine types. The vaccine type showed a strong effect on IgG levels, at the same time, this strong effect is "equally strong" for every genotype. We need to confirm this because if there is an interactive effect, we needed a stratified analysis. We slightly modified the sentence as follows:

“The vaccine type had a strong effect on antibody levels, and no interactive effect between rs671 and the type of vaccine was indicated (p for interaction = 0.5 in a model additionally includes the interactive term to model 3), indicating the same effect on every genotype”.

We also modified Line 209- (in the original manuscript) as follows (underlined): 

In our study, we included multiple covariates, such as smoking status, ethanol intake, exercise, perceived stress, steroid use, allergic disease, and dyslipidemia, to improve the validity and rigor of our statistical assumption based on previous research.

Reviewer 3 Report

1. Although the ALDH2 variant allele RS671 is associated with reduced response to anti-SARS-CoV-2 spike protein IgG vaccine, due to the low number of subjects used in the study, too many factors involved, insufficient discussion and unclear mechanism, the conclusion may not be universal. Assume that the conclusion is correct.Immunity to other vaccines should have the same effect.Therefore, either increase the number of subjects or test for changes in immune antibodies from more than two vaccines.

2. The materials provided are missing Table S1 and Figure S1, and the url provided cannot be opened.

Author Response

Thank you very much for your time and comments.

We agreed that our study has important limitations as you pointed out. We discussed:

  • the cohort-size problem
  • the effort to adjust possible confounders
  • a lack of a mechanism
  • generalizability

in Discussion, also, referred to the fundamental issue that was raised here. We need to examine our hypothesis for different types of vaccines.

Round 2

Reviewer 2 Report

The authors improved the manuscript, and the changes made to the text reflect my recommendations. Therefore, I believe the manuscript can now be accepted for publication.